# Intrauterine Growth Restriction Due to Gestational Diabetes: From Pathophysiology to Diagnosis and Management

**DOI:** 10.3390/medicina59061139

**Published:** 2023-06-13

**Authors:** Zacharias Fasoulakis, Antonios Koutras, Panos Antsaklis, Marianna Theodora, Asimina Valsamaki, George Daskalakis, Emmanuel N. Kontomanolis

**Affiliations:** 1Department of Obstetrics and Gynecology, General Hospital of Athens ‘Alexandra’, National and Kapodistrian University of Athens, Lourou and Vasilissis Sofias Ave, 11528 Athens, Greece; 2Department of Internal Medicine, General Hospital of Larisa, Tsakalof 1, 41221 Larisa, Greece; 3Department of Obstetrics and Gynecology, Democritus University of Thrace, 6th km Alexandroupolis-Makris, 68100 Alexandroupolis, Greece

**Keywords:** IUGR, gestational diabetes, adverse perinatal outcome, ultrasonographic monitoring

## Abstract

Intrauterine growth restriction (IUGR) represents a condition where the fetal weight is less than the 10th percentile for gestational age, or the estimated fetal weight is lower than expected based on gestational age. IUGR can be caused by various factors such as maternal, placental or fetal factors and can lead to various complications for both the fetus and the mother, including fetal distress, stillbirth, preterm delivery, and maternal hypertension. Women with gestational diabetes are at an increased risk of developing IUGR. This article reviews the different aspects of gestational diabetes in addition to IUGR, the diagnostic methods available for IUGR detection, including ultrasound and Doppler studies, discusses the management strategies for women with IUGR and gestational diabetes and analyzes the importance of early detection and timely intervention to improve pregnancy outcomes.

## 1. Introduction

Fetal growth results from the intricate interplay between the fetus’s genetic capacity for growth and the effects of the intrauterine environment on the mother. The three leading sources of intrauterine growth restriction (IUGR) are placental, maternal, and fetal. Although these components frequently overlap, such a separation is more theoretical than practical. Low birth weight puts the child at risk of cardiovascular diseases and type 2 diabetes mellitus, in addition to the increased risk of short-term problems caused by impaired fetal growth, with intrauterine fetal mortality as the most serious [1,2]. IUGR is discovered when the approximated weight of the fetus based on an ultrasound falls under the 10th percentile at the gestational period. When IUGR is diagnosed, a fetal weight is low because of a problem with pathologic growth restrictions. Small for gestational age (SGA) is a phrase that describes a fetus whose approximated weight falls under the 10th percentile. Nevertheless, SGA fails to represent that the fetus has a disease that causes it to be small for its age. About 70% of fetuses with SGA are congenitally small, which means they are healthy though small. The rest are fetuses with IUGR, which puts them at a greater risk of difficulties during birth and means they need close observation [3,4,5,6].

In diabetes pregnancy, fetal growth anomalies are common, with large for gestational age (LGA) as the commonest. LGA is extensively researched and is particularly commonly related to maternal hyperglycemia. Nevertheless, other variables such as prenatal weight gain and maternal lipids could also have an impact [7,8,9,10,11]. Diabetic pregnancy, on the other hand, can be linked to decreased fetal development. Long-term and poorly managed type 1 diabetes is related to an extensive range of clinical issues, most related to diabetic vasculopathy. Because the placenta is entirely made up of arteries, maternal vasculopathy has been associated with placental dysfunction and resultant fetal development limitation [12,13,14]. This review examines the diagnosis, pathophysiology, and therapy of IUGR related to diabetics.

## 2. Pathophysiology of Fetal Growth Restriction Due to Gestational Diabetes

The pathophysiology of fetal growth in the context of gestational diabetes mellitus is intricate and multifaceted. Gestational diabetes, characterized by glucose intolerance during pregnancy, can significantly impact fetal development, and has been associated with several fetal complications, including macrosomia and intrauterine growth restriction (IUGR) [13]. Many risk factors have been identified; however, the main pathophysiology of gestational diabetes is still under research (Figure 1). The underlying mechanism of fetal growth abnormalities in Gestational Diabetes Melitus (GDM) is often referred to as “Pedersen’s Hypothesis” [15]. According to this theory, maternal hyperglycemia leads to an increase in the transfer of glucose to the fetus via the placenta. This excess glucose stimulates the fetal pancreas to produce more insulin, which acts as a growth hormone, leading to macrosomia [16].

While macrosomia is a common outcome, gestational diabetes can also lead to IUGR. One of the mechanisms proposed for IUGR involves maternal microangiopathy—damage to small blood vessels—which is associated with longstanding diabetes [17]. Microangiopathy can impair the placenta’s ability to supply adequate nutrients and oxygen to the fetus, leading to growth restriction [18]. Additionally, changes in lipid metabolism have been observed in gestational diabetes, with increased maternal serum free fatty acids (FFAs), which can cross the placenta and contribute to excess fetal growth [19]. Furthermore, elevated levels of certain inflammatory markers, such as tumor necrosis factor-alpha (TNF-α), have been implicated in the pathophysiology of gestational diabetes and associated fetal growth abnormalities [20].

Maternal microvascular complications, such as microangiopathy, can lead to a decrease in uteroplacental blood flow, resulting in fetal hypoxia and nutrient deprivation [21]. These conditions can activate a series of metabolic and hormonal changes in the fetus, eventually leading to growth restriction. Furthermore, it has been observed that maternal hyperglycemia can lead to placental oxidative stress, which could induce changes in the placental structure and function, further contributing to IUGR [22].

In addition to glucose, other metabolic substrates such as amino acids are essential for fetal growth. Altered amino acid metabolism, as seen in gestational diabetes, can contribute to deviations in fetal growth. Gestational diabetes-associated changes in the maternal and fetal amino acid profile, specifically, reduced levels of essential amino acids, can lead to fetal growth restriction [23].

It is important to note that gestational diabetes-associated fetal growth abnormalities can have long-term health implications for the offspring, including an increased risk of obesity and type 2 diabetes in later life [24]. This underlines the importance of early detection, strict glycemic control, and appropriate management of gestational diabetes to optimize both immediate and long-term health outcomes for mother and child.

## 3. Animal Studies

Fetal growth is stunted when an unhealthy intrauterine milieu prevents the fetus from receiving enough of the nutrients it needs to grow. In one rat model of UPI, the pregnant rats (Sprague Dawley rats) have the bilateral uterine artery ligation surgery on day 18 where the term is 22, of their pregnancies and then give birth naturally. Lowered levels of insulin, glucose, IGF-1, oxygen, and amino acids are observed in uteroplacental insufficiency (UPI) fetal rats, mirroring the metabolic profile of human babies with a developmental delay [25,26]. Until about 7 weeks old, UPI progeny had significantly lower birth weights compared to controls, but eventually caught up. UPI progeny weights were higher than control weights at 26 weeks old, with the fastest growth occurring between 7 and 10 weeks old [27,28]. At the one-week mark, neither insulin nor blood glucose levels had changed noticeably. However, by 7–10 weeks old, IUGR rats showed signs of mild fasting hyperinsulinemia and hyperglycemia. At 15 weeks, the UPI male offspring were resistant to insulins and intolerant to glucose. Around 7 weeks, before the onset of hyperglycemia, UPI males already had a reduced capacity for the first stage of insulin release in response to glucose. Beta-cell mass, islet size, and pancreatic function did not differ significantly between infants aged 1 week and 7 weeks. Male infants born to women with UPI had a relatively lower beta-cell mass than the control group. After the loss of beta-cell mass, Pdx-1 expression went down as well. By 6 months, the UPI progeny had developed diabetes, which is similar to the type 2 diabetes observed in adults and is caused by a progressive lack of insulin synthesis and insulin activity [27]. Several investigations have found that IUGR is associated with increased oxidative stress in the developing fetus [25,26].

Mitochondrial reprogramming could be an essential mechanism that lets the developing fetus survive in a low-energy setting, but it can have negative effects on cells that have higher energy demands. Islets from children exposed to UPI showed steady increases in reactive oxygen species production and oxidative stress [18]. ATP production is inhibited and electron transport chain complexes I and II show declining activity in UPI islets. As a result, insulin production was attenuated in UPI islets, where mitochondrial dysfunction and mitochondrially encoded genes were less likely to be activated. Studies using UPI models suggest that an aberrant intrauterine milieu has an irreversible effect on insulin signaling, which results in greater gluconeogenesis [18,27,28,29]. An increased basal rate of hepatic glucose production (HGP) and a reduction of the degree to which HGP was regulated by insulin in male offspring exposed to UPI began to become apparent between weeks 7 and 9. There was a threefold increase in glucose-6-phosphate mRNA and phosphoenolpyruvate carboxykinase (PEPCK) levels, and a significant decrease in Akt-2 phosphorylation and IRS2 in UPI descendants [30]. These occurrences occurred early in adulthood for UPI kids, before hyperglycemia manifested itself, proving that UPI causes a primary impairment in hepatic metabolism, which in turn causes overt hyperglycemia due to beta-cell abnormalities.

## 4. Histologic Findings

A diabetic background can cause a variety of histopathological abnormalities in the placenta. While some are not unique to diabetic placentas, villus maldevelopment, fetal vasculitis, and vascular malperfusion may be regarded as risk factors for fetal growth restriction. Maternal vascular malperfusion develops due to insufficient and partial trophoblast infiltration and poor spiral artery transformation [30,31]. Furthermore, un-remodeled uterine arteries can exhibit characteristics of decidual vasculopathy. Previously, it was discovered that about a quarter of type 1 diabetic placentas had decidual vasculopathy [32,33].

It is well-established that preeclampsia occurs at a much higher rate in pregnancies where the mother has type 1 diabetes. The risk also increases over time, with the highest occurrence being observed in females with diabetic nephropathy, who are thus at the highest possible chance of developing IUGR [27]. In terms of histology, distal villus maldevelopment includes conditions including distal villous hypoplasia, dysmorphic villi, immaturity, and chorangiosis. In many cases of intrauterine growth restriction (IUGR), the placenta shows these changes, which are linked to a decline in the amount of villus surface area that is actively producing trophic factors [28,29]. Diabetic placentas frequently show fetal vasculitis, which may be an indication of the fetal inflammatory response. There is debate over the role of fetal vasculitis in IUGR, but some research has found a correlation between inflammatory markers and IUGR [30,31].

## 5. Serological Markers

Currently, gestational diabetes mellitus can be identified by specific serological markers (Table 1). However, more and more studies focus on early detection and prognosis of the disease. Several serological markers have been identified that can help detect gestational diabetes at an early stage, monitor its progression, and guide treatment.

One such marker of primary interest is hemoglobin A1c (HbA1c), a form of hemoglobin that is chemically linked to a sugar, typically glucose. The higher the level of glucose in the blood, the more HbA1c is formed. As such, it provides an accurate estimate of average blood glucose levels over the past 2–3 months. While HbA1c is traditionally used in diagnosing and monitoring type 1 and type 2 diabetes, its role in gestational diabetes is currently a subject of extensive research [34].

Another serological marker related to gestational diabetes is insulin. Pregnant women often exhibit some degree of insulin resistance due to the hormone changes that occur during pregnancy. However, when the body cannot compensate for this resistance by producing enough insulin, gestational diabetes can develop. Monitoring insulin levels, therefore, can be crucial in the early detection and management of gestational diabetes [35].

Apart from these, there are several other biochemical markers associated with inflammation, oxidative stress, and endothelial dysfunction that have been linked to gestational diabetes. These include, but are not limited to, C-reactive protein (CRP), interleukin-6 (IL-6), tumor necrosis factor-alpha (TNF-α), and adiponectin. Elevated levels of these markers suggest an inflammatory response, which can contribute to insulin resistance and the development of gestational diabetes [36,37,38].

Finally, emerging research has pointed to the potential role of certain gut microbiota-derived metabolites, such as trimethylamine N-oxide (TMAO), as serological markers for gestational diabetes. Recent studies have shown that elevated TMAO levels are associated with an increased risk of gestational diabetes, suggesting that gut microbiota dysbiosis may play a role in the disease’s pathogenesis [39].

Except for early diagnosis of gestational diabetes, an increasing body of evidence suggests the use of important serological markers has proven to be an invaluable tool in identifying IUGR.

One key marker that has been identified is advanced glycation end products (AGEs). These compounds are typically formed when proteins or lipids become glycated after exposure to sugars, and they can be particularly harmful in the context of diabetes, where blood glucose levels are often poorly controlled. AGEs can interfere with normal cellular function and induce inflammation, leading to endothelial dysfunction and vascular complications. Studies have shown that high levels of AGEs are often present in women with gestational diabetes, signifying the occurrence of microangiopathy [40].

Another crucial serological marker in the context of gestational diabetes and IUGR is the vascular endothelial growth factor (VEGF). This protein plays a pivotal role in blood vessel formation, a process that is crucial to the development and function of the placenta. Alterations in VEGF levels have been associated with many pregnancy complications, including IUGR. In cases of gestational diabetes, elevated blood glucose levels can cause overexpression of VEGF, leading to abnormal placental vasculature and impaired nutrient and oxygen transport to the fetus, thereby inducing IUGR [41].

Inflammatory cytokines, such as tumor necrosis factor-alpha (TNF-α) and interleukin-6 (IL-6), are also important serological markers to consider. These substances are typically elevated in response to stress and can contribute to inflammation and endothelial dysfunction, further complicating the picture of gestational diabetes and IUGR [42,43].

## 6. Ultrasonographic Evidence

Although placental histopathology observations give significant and clinically reliable information, they are always comprehensive and reflect pathogenic mechanisms and alterations that occurred weeks or months prior to delivery. Doppler ultrasound, which allows for real-time evaluation of placental and fetal circulation, is commonly employed in the management of high-risk pregnancies. Furthermore, irregular uterine and umbilical Doppler velocity waveforms are linked to abnormal placental pathology in IUGR [44,45,46,47]. Maintaining healthy fetal growth requires effective fetoplacental and uteroplacental circulation.

Pietryga et al. found that pregnant women having type 1 diabetes who had vasculopathy had considerably higher uterine artery vascular impedance. Additionally, rising uterine artery scores were linked to a higher frequency of SGA babies. White found that aberrant uterine Doppler was nearly entirely detected in females with R/F retinopathy and nephropathy in this investigation [48,49]. Doppler findings were mostly adequate in females with simple diabetes or isolated retinopathy. Umbilical artery aberrant flow was far less common and unrelated to vasculopathy. This demonstrates that changes in the umbilical artery may develop slowly as an adaptation to low uteroplacental perfusion, and that malfunction in uteroplacental circulation is a fundamental disease in the placentas of females with significant vasculopathy.

Salvesen et al. found that diabetic vasculopathy is directly related to impaired placental functioning. It involved 41 females with type 1 diabetes that had cordocentesis within 24 h of giving birth. Researchers discovered that all fetuses of mothers with hypertension and nephropathy had an umbilical vein pH significantly below the normal gestation mean (5th percentile) [49].

## 7. Diagnosis

Generally, IUGR fetuses are distinguished by an approximated fetal weight (EFW) lower than the 10th percentile. However, the Royal College of Obstetricians and Gynecologists (RCOG) indicates that IUGR diagnosis may be determined by means of the fetal abdominal circumference (AC) that is below the 10th percentile [3,4,5]. As such, the correct dating of pregnancy is crucial for the IUGR diagnosis with an assessment of crown-rump length (CRL), ideally between the gestation period of 9 and 13 weeks, presenting the accurate foundation for more examination of fetal development and needs to be performed within these weeks.

## 8. Management

Monitoring pregnant women with gestational diabetes mellitus (GDM) is a multi-faceted process that requires regular assessment of both maternal and fetal health parameters. Low APGAR scores, intrauterine fetal death (IUFD), neonatal mortality, preterm delivery, and impaired neurodevelopment of the child are all risks of IUGR of serious pregnancy results. The goal of careful monitoring of the fetus with IUGR is to determine the most risk-free timing for delivery. Even the most skilled obstetrician may struggle with this, particularly in instances of fetal preterm birth. As a result, various considerations must be carefully considered when deciding on the date and style of delivery for an IUGR fetus. The etiology of IUGR is always essential in terms of management. Determining the underlying cause of restricted fetal development is critical, and placental inadequacy is among the most significant challenges to monitoring unexpected types of IUGR, particularly in females with preeclampsia and nephropathy.

The primary goal of monitoring is to maintain blood glucose levels within a specific target range to minimize the risks of GDM-related complications [50]. Blood glucose monitoring is the cornerstone of GDM management. Self-monitoring of blood glucose (SMBG) is recommended to assess the effect of dietary modifications and physical activity on blood glucose levels and to guide insulin therapy if required [51]. The American Diabetes Association (ADA) recommends SMBG at least four times daily: fasting, either before meals or 1–2 h postprandially [52]. HbA1c testing may also be used. In addition to glucose control, regular antenatal visits are recommended to monitor maternal blood pressure and weight gain, and screening for urinary tract infections, which are more common in women with GDM.

The decision to deliver is frequently based on proof of deterioration of the maternal condition and acute fetal distress or both. Adequate prenatal surveillance and timely delivery could minimize the likelihood of perinatal morbidity and mortality in diabetic women. Furthermore, because of the unanticipated delivery time, the RCOG advises that a single dose of corticosteroids be administered during pregnancy with restricted fetal development and expected preterm delivery until 35 + 6 weeks of gestation [4]. Aside from food and glycemic control, which are critical for a positive outcome, there are various techniques for fetal surveillance listed below, although only a few have been shown to determine neonatal results.

Diabetes and IUGR-complicated pregnancies among women must be handled by maternal–fetal medical professionals competent in diabetic pregnancy within a perinatal facility with accessibility to a neonatal intensive care unit.

Women with different types of diabetes may experience changes in their fetus’s growth patterns, making serial ultrasound measurements all the more important. It is now up to clinical discretion to decide how long to wait between scans because no optimal intervals have been identified. The risk of a false-positive diagnosis can be reduced, however, by taking further measurements at least every three weeks. Fewer gaps in fetal biometry monitoring may be beneficial for women with IUGR and type 1 diabetes [53].

Ultrasound imaging during pregnancy typically includes an evaluation of the fetus’s AFV (amniotic fluid volume). Although oligohydramnios has been associated with perinatal mortality and SGA, its ability to predict an individual’s risk of these outcomes is limited. Thus, AFV evaluation is not sufficient for monitoring IUGR fetuses alone. Nevertheless, when combined with Doppler ultrasounds or included in biophysical profiles, this may offer additional information on fetal well-being that may be utilized for therapeutic purposes (BPP). The amniotic fluid volume (AFV) can be calculated in two ways: subjectively (based on the examiner’s visual judgment) and objectively (using the amniotic fluid index [AFI] and the single deepest pocket [SDP]). When subjective assessments reveal a lower AFV or when patients are at rising risk of IUGR, objective AFV assessment is encouraged. A contrast of SDP and AFI found that AFI was related to a greater rate of erroneous positive oligohydramnios diagnosis, which might also lead to unwarranted induction of labor. As a result, SDP assessment could be preferable to AFI, particularly in preterm pregnancies [54].

Doppler ultrasound accurately reveals impairments of fetoplacental and uteroplacental circulation associated with maternal vasculopathy in the event of IUGR. Umbilical Doppler velocimetry is extensively suggested as a primary tool for the IUGR fetus follow-up. Additionally, it is critical in diagnosing the reason for restricted fetal development (constitutional SGA/other causes vs. placental insufficiency). Women having type 1 diabetes who are at risk of IUGR, particularly the ones with diabetic nephropathy, might benefit from starting low-dose aspirin at 12 weeks of gestation. Delivery needs to be considered when Ductus Venosus (DV) doppler anomalies or Umbilical Vein (UV) pulsation are observed in fetuses with lacking or reverted end-diastolic velocity detectable before 32 weeks. Antenatal steroids need to be given until 35 + 6 weeks of gestation if preterm delivery is expected [55].

## 9. Conclusions

Gestational diabetes mellitus and its correlation with intrauterine growth restriction present a complex interplay of physiological, metabolic, and molecular factors. This review illuminates the multifactorial pathophysiology, underlining the role of hyperglycemia, microangiopathy, placental abnormalities, and altered nutrient transport. Further, it emphasizes the pivotal role of early diagnosis of gestational diabetes in preventing the onset of IUGR and other adverse pregnancy outcomes. Current diagnostic approaches, including the use of serological markers and ultrasonography, have been discussed, highlighting their efficacy and limitations. The management of gestational diabetes-induced IUGR involves maintaining optimal blood glucose levels, regular fetal monitoring, and timely delivery to avoid further complications. The importance of a comprehensive approach, including lifestyle modifications and, if necessary, medication, has been underscored. Despite significant advancements in understanding the pathophysiology and management of gestational diabetes and IUGR, there remain several areas that warrant further research. These include the identification of novel serological markers for the early detection of gestational diabetes, a deeper understanding of placental abnormalities, and the development of more effective therapeutic strategies. The goal is to improve both immediate and long-term health outcomes for mother and child. The complexity of gestational diabetes and its impact on fetal growth necessitates ongoing, interdisciplinary collaboration and research efforts to ensure the best possible outcomes for these pregnancies.

## Figures and Tables

**Figure 1 medicina-59-01139-f001:**
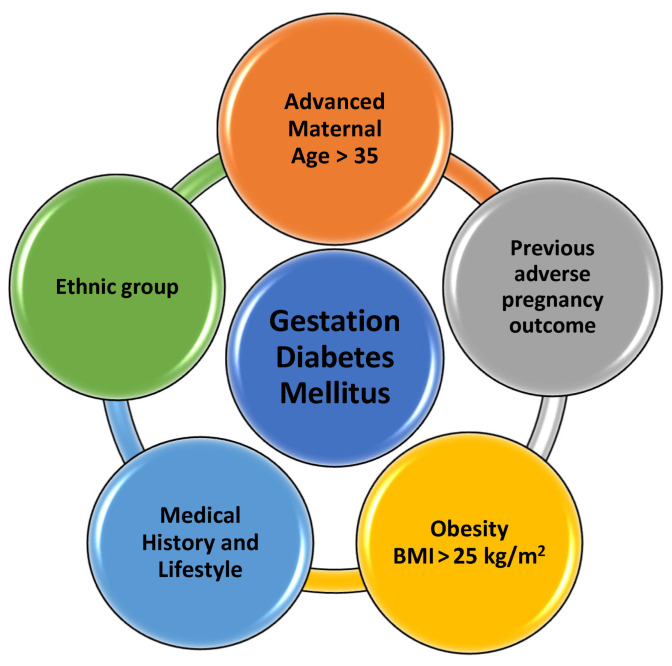
Risk factors associated with gestational diabetes mellitus. Advanced maternal age > 35, previous adverse pregnancy outcome, overweight and obesity (BMI > 25 and 30 kg/m^2^ subsequently), previous history of gestational diabetes or family history, and specific ethnic groups (Asian, Hispanic, African, Native American, Pacific islander, Indigenous Australian) are all risk factors associated with gestational diabetes mellitus. (BMI, body mass index) [12,13,14,15,16].

**Table 1 medicina-59-01139-t001:** Diagnostic markers for gestational diabetes mellitus [34,35,36].

Predictor	Pregnancy Period	Values
Impaired glucose tolerance	Between weeks 24 and 28	Assessed by oral glucose tolerance tests (OGTT): fasting > 5.1 mmol/L, 1st hour > 10 mmol/L or 2nd hour > 8.5 mmol/L
Glycated hemoglobin	-	>6.5% (48 mmol/mol)
Body mass index	-	>30 kg/m^2^

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
