# Peer review of "Intrauterine Growth Restriction Due to Gestational Diabetes: From Pathophysiology to Diagnosis and Management"

_medicina, 2023, doi:10.3390/medicina59061139_

Round 1

Reviewer 1 Report

The authors report on an interesting and attractive subject: the intauterine (fetal) growth restriction (IUGR) in patients associating gestational diabetes.

 Major comments:

1. If the title is “Diagnosis and management of…”, I consider that the “Pathophysiology of Diabetes in pregnancy” and “Animal studies” could have been omitted.

Also, although “Histologic findings” are important characteristics of defective placental-fetal interface, they will add very little help to the clinician for establish the diagnosis and / or the management of a such high risk pregnancy.

Furthermore, it will be important for the clinician for attempting these purposes to measure the vascular cell adhesion molecule-1 (VCAM-1)?

 2. The „Ultrasonographic evidence” is the strong part of the article.

 3. The “Management” should be much more strictly focused on the association of SGA and/or IUGR with gestational diabetes and less with the general management in case of SGA or IUGR. I consider that would be more appropriate with the title of the manuscript and much more interesting for a clinician reader.

Author Response

Dear reviewer 1,

Thank you for your time on reviewing our manuscript and for your valuable comments.

1. It is true that with the given title, the article should focus only in diagnosis and management of IUGR due to gestation diabetes and when writing the article, we found interesting both the histologic findings and animal studies. However, with the previous title, both parts seem to be out of topic. Thus, in order to maintain them in the text, we made a change on the title of the article to  "Intrauterine Growth Restriction due to Gestational Diabetes: From Pathophysiology to Diagnosis and Management". Moreover, we have made major changes on the serological markers part, where now we analyze both gestational diabetes AND IUGR serological markers for early detection and diagnosis.

2. Thank you for your comment.

3. Gestational diabetes is a multifactor pathology that should be treated by a cooperation of endocrinologists and obstetricians. To maintain the article in the obstetricians scope, but in addition to your valuable recommendations, we added the role of glycemic control in “management section”.

Reviewer 2 Report

Diagnosis And Management Of Intrauterine Growth Restriction (IUGR) Associated With Gestational Diabetes

By Zacharias Fasoulakis, et al.

The paper elicits diagnosis/management and path physiology of IUGR with gestational diabetes. The article is of interest for audience as not substantial innovative suggestions can be assumed.

It seems more conceivable to widen the literature data to produce a state of the art or a metanalysis in the field.

Author Response

Dear reviewer ,

Thank you for your time on reviewing our manuscript and for your valuable comments. We have made great modifications in the paper and have now added more and recent citations.

Reviewer 3 Report

The manuscript concerns diagnosis and management of intrauterine growth restriction associated with pregestational diabetes. However, it is written so inaccurately, that even the title is misleading.

Gestational diabetes (as defined by ADA 2022, FIGO 2015 and WHO 2013 criteria) is not associated with IUGR.

There are a lot of improper references, especially after ref 17.

Ref. 16 includes several articles.

Many references are outdated. There are almost no references dated after 2014.

Many statements are not supported by a reference at all.

Overall, the review do not seem informative.

Some phrases are not clear due to English misuse (ex., « Male infants born to women with UPI had a relative beta-cell mass that was 50% of the controls at 15 weeks and less than 33% of controls at 26 weeks old» when the authors mean male offspring of rats with UPI).

Author Response

Dear reviewer,

Thank you for your time on reviewing our manuscript and for your valuable comments.

References are all now correct and have been modified to meet the journal’s guidelines.

We have added new references and many of them are from the latest years.

Many corrections are made throughout the text so as to be more comprehensive and to support our main topic on gestational diabetes as a cause of IUGR.

Round 2

Reviewer 1 Report

The authors have done significant improve of the manuscript.

Reviewer 2 Report

Please to refer to the enclosed file. 
